# Identification of Candidate Genes for Root Traits Using Genotype–Phenotype Association Analysis of Near-Isogenic Lines in Hexaploid Wheat (*Triticum aestivum* L.)

**DOI:** 10.3390/ijms22073579

**Published:** 2021-03-30

**Authors:** Tanushree Halder, Hui Liu, Yinglong Chen, Guijun Yan, Kadambot H. M. Siddique

**Affiliations:** 1UWA School of Agriculture and Environment, The University of Western Australia, 35 Stirling Highway, Crawley, WA 6009, Australia; hui.liu@uwa.edu.au (H.L.); yinglong.chen@uwa.edu.au (Y.C.); guijun.yan@uwa.edu.au (G.Y.); 2The UWA Institute of Agriculture, The University of Western Australia, 35 Stirling Highway, Crawley, WA 6009, Australia; 3Department of Genetics and Plant Breeding, Faculty of Agriculture, Sher-e-Bangla Agricultural University, Dhaka 1207, Bangladesh

**Keywords:** root, wheat, near-isogenic lines, QTL, SNP, candidate genes, protein

## Abstract

Global wheat (*Triticum aestivum* L.) production is constrained by different biotic and abiotic stresses, which are increasing with climate change. An improved root system is essential for adaptability and sustainable wheat production. In this study, 10 pairs of near-isogenic lines (NILs)—targeting four genomic regions (GRs) on chromosome arms 4BS, 4BL, 4AS, and 7AL of hexaploid wheat—were used to phenotype root traits in a semi-hydroponic system. Seven of the 10 NIL pairs significantly differed between their isolines for 11 root traits. The NIL pairs targeting qDSI.4B.1 GR varied the most, followed by the NIL pair targeting qDT.4A.1 and QHtscc.ksu-7A GRs. For pairs 5–7 targeting qDT.4A.1 GR, pair 6 significantly differed in the most root traits. Of the 4 NIL pairs targeting qDSI.4B.1 GR, pairs 2 and 4 significantly differed in 3 and 4 root traits, respectively. Pairs 9 and 10 targeting QHtscc.ksu-7A GR significantly differed in 1 and 4 root traits, respectively. Using the wheat 90K Illumina iSelect array, we identified 15 putative candidate genes associated with different root traits in the contrasting isolines, in which two UDP-glycosyltransferase (UGT)-encoding genes, *TraesCS4A02G185300* and *TraesCS4A02G442700*, and a leucine-rich repeat receptor-like protein kinase (LRR-RLK)-encoding gene, *TraesCS4A02G330900*, also showed important functions for root trait control in other crops. This study characterized, for the first time, that these GRs control root traits in wheat, and identified candidate genes, although the candidate genes will need further confirmation and validation for marker-assisted wheat breeding.

## 1. Introduction

Wheat (*Triticum aestivum* L.) is an important cereal crop, which is cultivated worldwide and significantly for international food markets. The ever-changing global climate has increased abiotic and biotic stresses, constraining wheat production. Considerable attention has been paid to wheat root systems in recent years in an attempt to adapt to climate change and achieve sustainable yields. Roots are directly associated with soil, water, and nutrient uptake, anchoring plants in the substrate, interacting with plant microbes, and storing resources; therefore, root architecture greatly contributes to crop yield [1]. Wheat genotypes with modified root mass had higher water uptake than those with standard root mass, resulting in a 200 kg ha^−1^ yield advantage in drought conditions [2]. Under nutrient-deficit conditions in the soil, fast- and early-proliferating roots with long and dense root hairs enhanced macro- and micronutrient uptake in wheat; vigorous root growth helps in early nutrient acquisition [3]. Therefore, the root system is essential for improving stress tolerance and yield potential. Identifying genetic variation and genes responsible for root traits in wheat is of great interest.

Single-nucleotide polymorphism (SNP) is a powerful tool for studying diversity within the individual species at the genomic level. Due to its abundance, ubiquity, and amenability to high-throughput automation [4], SNP is now widely used in quantitative trait loci (QTL) mapping, marker-assisted selection (MAS), genome-wide association studies (GWAS), and genomic selection (GS) [5,6]. The wheat 90K Infinium iSelect SNP array has been used in many wheat studies; several grain-related genes and potential markers have been identified using this approach [7,8]. However, few studies have focused on identifying genes associated with root traits in wheat.

Rufo et al. (2020) found that higher genetic diversity in root traits is mostly associated with the A (46%) and B (48%) genomes compared with the D (6%) genome in wheat. They identified 31 candidate genes for root system architecture in landraces and modern wheat varieties using a GWAS study [9]. Two root trait architecture candidate genes, *AX*-*110092021* (encoding protein kinase) and *AX*-*95681859* (encoding E3 ubiquitin-protein ligase), were identified on chromosomes 5BL and 1BL in Chinese wheat cultivars [10]. Protein kinase genes were expressed stably on wheat roots at different stages, and played important roles in root anisotropic growth [11]. *VERNALIZATION1 (VRN1*), a MADS-box protein-encoding gene on chromosome 5B, is associated with root length variations at different wheat growth stages [12]. An earlier study reported 63 marker-trait associations with different root traits, including rooting depth (RD), root mass (RM), and root diameter (Rdia); the height-reducing gene *Rht* encoding a DELLA protein played an important role in increasing RD but decreasing RM and Rdia [13]. A malate transporter gene, *ALMT1* (encoding a novel aluminum (Al)-activated malate transport protein) on chromosome 4DL, is expressed in root apices and improves Al stress tolerance [14]. The recently available reference genome has opened an opportunity to identify trait-controlling genes in wheat, including their sequence and diversity in molecular breeding [15]. Due to difficulties in studying root systems, wheat root phenotyping studies are limited to early growth stage under controlled environments [16]. Potential genes for many root traits from pure lines are yet to be discovered for root improvement.

Near-isogenic lines (NILs) are pairs of genetically identical lines, except for target genetic loci, which are important genetic resources for identifying candidate genes and their functions associated with phenotypic traits. The NILs used in the present study were developed targeting different genomic regions (GRs) on wheat chromosome arms 4BS [17], 4AS [18], 4BL [19], and 7AL [20]. The NILs were developed using the heterogeneous inbred family (HIF) method [19], and therefore, the genetic backgrounds of each pair are different even if they target the same GR, which allowed comparisons among the NIL pairs to narrow down the candidates of the trait-linked markers and genes. In this study, we investigated root traits in 10 pairs of wheat NILs using the wheat 90K Infinium iSelect SNP array as a genotyping method. The study aimed to (1) characterize phenotypic variabilities between isolines for different root traits, (2) identify candidate genes associated with root traits contrasting in NIL pairs, and (3) investigate functional relationships between the identified candidate genes and previously reported genes on the respective GRs.

## 2. Results

### 2.1. Phenotypic Variation and Correlation of Root Traits

Multiple NIL pairs (Table 1) exhibited a wide range of variations in diverse root traits and a few shoot traits (Table 2; Appendix A). Except for pairs 1, 7, and 8, the NIL pairs significantly differed between isolines in the measured root traits. For pairs 1–4 targeting qDSI.4B.1 GR, pair 2 significantly differed in root diameter class length (<0.25 mm) (RDCL1), specific root length (SRL), and shoot height (SH); NIL2b had 35.69% more SRL than NIL2a. NIL2b also had higher rooting depth (RD) than NIL2a, although the difference was not significant (Table 2; Figure 1). Pair 3 significantly differed in root/shoot ratio (RSR) only. Pair 4 significantly differed in root surface area (>0.25 mm diameter class (RSA1) and <0.25 mm diameter class (RSA2)) and RDCL1 and RDCL2 (root diameter class length (>0.25 mm)). NIL4a showed higher (non-significant) RD than NIL4b (Table 2; Figure 1).

For pairs 5–7, pair 5 significantly differed in RSR only, while pair 6 significantly differed in the most traits (5 of 11). NIL6a had almost twice the total root length (RL), root mass (RM) (Appendix A), and RDCL1 compared with NIL6b. Pair 6 also significantly differed in RSA2 and leaf number per plant (NLP). NIL6b had 35.56% more SRL than NIL6a (Table 2). 

For Pairs 9 and 10, NIL9a had significantly higher RDCL1 than NIL9b, and NIL10a had significantly higher RL (Appendix A), root diameter (Rdia), and RDCL2 than NIL10b. However, NIL10b had significantly more nodal root number per plant (NNR) than NIL10a (Table 2).

Significant correlation was found between different root and shoot traits of the NIL pairs. According to the correlation coefficient values, strong (r ≥ 0.7) and weak (r ≤ 0.3) correlations were denoted [24]. Correlation coefficient values between 0.3 and 0.7 were denoted as moderate correlation. Total root length showed strong positive correlation with RM, RSA2, RDCL1, and NLP; moderate positive correlation with RD, RSR, NNR, and SH; and moderate but negative correlation with Rdia. Root mass showed strong positive correlation with RD, RDCL1, and NNR, and moderate positive correlation with RSR, RSA2, SH, and NLP. Strong positive correlation was found between RD and NNR, whereas moderate but negative correlation was found between RD and Rdia. Other traits, such as RSA1 and RDCL2, RSA2 and RDCL1, RSA2 and NLP, and RDCL1 and NLP, also showed strong positive correlations (Appendix A). 

### 2.2. SNPs and Candidate Genes

Of all the contrasting NIL pairs, a total of 67 SNPs on the targeted chromosomes were identified as candidate tightly linked markers for the studied root traits; for each investigated GR, they showed common polymorphism between the isolines among all the pairs targeting the same GR (Appendix A); specifically, pairs 1–4, 5–7, and 9–10 showed polymorphism for 11, 50, and 6 SNPs, respectively (Appendix A). In pairs 1–4, 5–7, and 9–10, 5, 11 and 2 SNPs showed polymorphism within the respective GRs. Using those 67 polymorphic SNP markers, from the nine NIL pairs, a total of 181 candidate genes were identified (Appendix A). However, only those genes with functions that have been previously reported to be responsible for root traits in wheat and other crops (especially cereal crops) were considered as important candidates (Table 3). Fifteen candidate genes tightly linked with the SNPs were identified (Table 3). On GR qDSI.4B.1, *TraesCS4B01G114500* and *TraesCS4B01G114800*, associated with the marker Excalibur_c869_2091 (Figure 2), were the genes annotated to encode the Phox-associated domain and bHLH-MYC and R2R3-MYB transcription factors N-terminal proteins, respectively (Table 3).

For pairs 5–7, the *TraesCS4A02G163700* gene encoded the inositol-1-monophosphatase family protein. An LRR-RLK-encoding gene, *TraesCS4A02G330900*, was located very close (7.78 kb away) to the marker BobWhite_c27287_232 (Table 3; Figure 3). The gene *TraesCS4A02G331000* was also linked (17.30 kb away) to BobWhite_c27287_232. Two genes, *TraesCS4A02G215800* and *TraesCS4A02G215700*, were 202.63 and 214.39 kb away from BS00098868_51, respectively (Table 3; Figure 3); these genes encode the F-box family protein. The UDP-glucoronosyl and/or UDP-glucosyl transferase-encoding gene *TraesCS4A02G185300* was associated with BS00094406_51 (962.14 kb away), and *TraesCS4A02G442700* was associated with CAP12_c3789_390 (11.58 kb away) and RAC875_c51781_771 (10.29 kb away) (Table 3). *TraesCS4A02G323500* and *TraesCS4A02G323400* were closely associated with Tdurum_contig12899_342 (5.86 and 10.11 kb away, respectively), and *TraesCS4A02G000300* was closely associated with Jagger_c9935_144 (90.30 kb away), encoding serine–threonine/tyrosine- protein kinase (Table 3; Figure 3).

For pairs 9–10, two genes linked to SNP CAP8_c3496_118, namely, *TraesCS7A02G431600* and *TraesCS7A02G431500*, had root-trait-related functions as F-box protein and serine–threonine/tyrosine- protein kinase, respectively (Table 3; Figure 4).

## 3. Discussion

### 3.1. NIL Pairs Show Different Phenotypic Performances


Among the 10 NIL pairs, 7 pairs showed significant differences between the isolines for the studied root traits. The HIF method for NIL development allowed the isolines of a NIL pair to be homozygous apart from the target locus [20]. Therefore, the isolines showed contrasting performances, suggesting that they contain the gene(s) controlling the corresponding root traits. NIL pairs that were developed from the same parents showed different phenotypic performances among the pairs. Due to the use of the HIF method in NIL development, each NIL pair was from a single seed descent (SSD) of F2 individuals, and for that reason [17], the NIL pairs had completely different genetic backgrounds from each other. Therefore, despite having the same parents, individual pairs showed different root phenotypic performances. By comparison, among all the pairs targeting the same GR, we narrowed down in each GR the candidate SNPs that could be linked to the root traits.

### 3.2. Targeted Genomic Regions of NILs Overlap Previously Reported QTL for Root Traits and Other Yield-Related Traits

The NILs used in this study to investigate root traits were initially developed to target different stress tolerance QTL (Table 1). Some of the QTL overlapped previously reported QTL for root or yield-related traits. Under drought stress, rooting depth QTL (qMRL.4B.1), root mass QTL (qTRB.4B.1), and grain yield QTL (qMRL.4B.2)—identified through linkage mapping of the hexaploid wheat population C306 × WL711—overlapped with qDSI.4B.1 in the current study [21]. Under well-watered and low-nitrogen (N) conditions, two root mass QTL (physical position, 75.74–222.28 Mb) were reported in the Xiaoyan 54 × Jing 411 population [26], which are very close to qDSI.4B.1 (59.61–75.74 Mb). From the meta-QTL analysis in durum wheat, Root_MQTL_52 (35.7–74.7 Mb) controlling rooting depth, root volume, root mass, root angle, root number, and root/shoot ratio overlapped the 4BS QTL [27]. Root_MQTL_51 for total root length overlapped the same QTL [27]. A QTL (QRv.sau-4B) controlling root volume was reported on chromosome 4BS (105.15 Mb) in bread wheat, only 29.4 Mb away from our studied qDSI.4B.1 [28]. Besides the root trait QTL, a few other yield-related QTL, such as QTL for thousand-kernel weight, were also found on chromosome 4BS as reported earlier [29,30].

Yield-related trait (including grain weight per spike, floret number per spikelet, spike length, and grain number per spike) QTL (flanking markers: Xwmc491–Xwmc96; 94.02–104.10 Mb) in an earlier study of wheat were very close to QSL.caas-4AS (0.14–94.02 Mb) [31,32]. Two QTL for wheat stripe rust resistance were collocated with Xgwm397, one of the flanking markers of qDT.4A.1 (Table 1), suggesting that this region might also be responsible for disease resistance [33].

Overlapping of QPhs.ocs-4B.1 with previously reported QTL was also revealed. Under low N, a QTL (QRDW.caas-4BL) controlling root mass identified in the Yangmai 16 × Zhongmai 895 population overlapped QPhs.ocs-4B.1; in the same population, a QTL for root mass under high N was just 19 Mb away from QPhs.ocs-4B.1 [34]. A root mass QTL (qRDW.LP-4B) under low phosphorus (P) identified in the Xiaoyan 54 × Jing 411 population also overlapped our targeted QTL [26]. 

A meta-QTL, Root_MQTL_86 for root mass, overlapped QHtscc.ksu-7A targeted in the current study, and another meta-QTL, Root_MQTL_85, was 10 Mb away from it [27]. Other grain-related QTL in cereals overlapped marker Xbarc121, a flanking marker of QHtscc.ksu-7A. Several QTL controlling thousand-grain weight and grain number in bread wheat for grain weight per plant in durum wheat, panicle number, seed setting rate, and grain weight per plant in rice also overlapped Xbarc121 and therefore QHtscc.ksu-7A [35,36]. 

The overlapping of our targeted QTL with other reported QTL for root, stress tolerance, and yield-related traits indicates that the genes underlying these GRs might have multiple functions in controlling root, yield, and stress tolerance traits in wheat. Therefore, characterization of these GRs might lead to the identification of important candidate genes for the traits.

### 3.3. Putative Candidate Genes Controlling Wheat Root Traits

The identified candidate genes reportedly have functional similarities for controlling root traits in different crops, including wheat (Table 4). The most outstanding candidates were the UDP-glycosyltransferase (UGT)-encoding genes *TraesCS4A02G185300* and *TraesCS4A02G442700* and the LRR-RLK-encoding gene *TraesCS4A02G330900*. UGT has the potential to improve crop root growth and development through the glycosylation of different hormones, including auxin, cytokinin. and abscisic acid [37]. Lignification of a crop cell is linked to monolignol glycosylation and co-expressed by the UGT gene, which had more lateral root formation and higher primary root elongation than the wild type despite the application of a strong root growth inhibitor (cytokinin) [38]. Overexpression of the UGT-encoding gene *OsIAGT1* reduced root length due to the upregulation of auxin synthesis genes and decreased endogenous indole acetic acid in rice seedlings [39]. Like UGT, LRR-RLK plays a crucial role in root growth and development, but there are no reports on the LRR-RLK gene controlling root traits in wheat. However, the LRR-RLK genes *TaLRRKs* are expressed highly in wheat roots [40]. Homozygous T-DNA insertion lines of *Arabidopsis* developed for LRR-RLK expression demonstrated root growth control through cytokinin mediation [41]; in other cases, a mutant with two T-DNA insertions (*BAK1/SERK3* and *IRK* genes) had inhibited root growth and *rlk902* had reduced root length [41]. Transgenic *Arabidopsis* developed from overexpression of a somatic embryogenesis receptor-like kinase (*SERK5*-Ler), an LRR-RLK, had improved rooting depth [42]. Mutant rice with loss of function of an LRR-RLK-encoding gene, *defective in outer cell layer specification 1* (*DOCS1*), had modified root gravitropism that resulted in a more open root cone angle (angle between most right and left of two external roots) and more radial root system, but reduced relative root elongation rate and shorter root hairs than the wild type [41]. Therefore, candidate genes encoding UGT and LRR-RLK could improve root traits in wheat, especially total RL and SRL. In addition, as the development of the 4AS NILs targeted drought tolerance QTL (Table 1), the genomic region might play a role in drought tolerance through improved root traits.

*TraesCS4B01G114500* identified in qDSI.4B.1 encodes the Phox-associated domain. There are few reports on the role of this protein in controlling root traits in wheat. Lee et al. (2008) found that the homolog of the protein phosphatidylinositol 3-phosphate (PtdIns(3)P) regulated root hair growth in *Arabidopsis* [43]. The protein also controlled root growth and development and lateral root length in *Arabidopsis* [44].

*TraesCS4B01G114800* identified in pairs 1–4 encodes bHLH-MYC and R2R3-MYB transcription factors N-terminal, an important protein for crop growth and development and abiotic stress tolerance in different crops, including wheat [45]. The regulatory genes encoding the proteins are poorly investigated in wheat [46]. MYC2 is essential for controlling root phenotype, and jasmonate-associated MYC2-LIKE (JAM) transcription factors inhibited root growth in *Arabidopsis* [47]. A cluster of MYC2, MYC3, and MYC4 played a role in root growth inhibition of *Arabidopsis* [48]. Wheat genes that encode MYC (*TaMyc*-B1) improved drought and salinity stress tolerance [49]. In wheat, apart from *TaMyc1* and *TaMyc*-B1, MYC genes have not been explored [49]. Thus, there is an excellent scope to study the mechanism of *TraesCS4B01G114800* in root trait control and drought stress tolerance in the future.

MYB transcription factor (TF) encoding genes in wheat, such as *TaPHR3-A1* and *TaSIM*, improved P uptake, salt stress tolerance, grain number, and notably root length in *Arabidopsis* [35,50]. A mutant of soybean overexpressing wheat R2R3-MYB TF (*GmMYB84*) had greater primary rooting depth at optimum reactive oxygen species (ROS) level and drought tolerance [51]. Under low phosphate conditions, overexpression of the rice gene *OsMYB2P-1* in *Arabidopsis* had greater lateral root density and longer primary and adventitious roots than the wild type [52]. In contrast, MYB TF *AftMYB93* negatively regulated lateral root development and played a role in root hair growth in *Arabidopsis* [53]. The sesame MYB gene *SiMYB75*, a root-specific gene, promoted root growth and abiotic stress (drought and salt) tolerance in *Arabidopsis* and sesame [54]. In this study, it was shown that *TraesCS4B01G114800* may play a role in RSR, RSA1, RSA2, and SRL, where the NIL isolines significantly differed in the traits.

Three of the candidate genes (*TraesCS4A02G331000*, *TraesCS4A02G215800*, and *TraesCS4A02G215700*) identified in pairs 5–7 and one gene (*TraesCS7A02G431600)* identified in pairs 9–10 encode F-box protein. Among the subunit proteins of Skp1–Cullin–F-box (SCF), F-box protein plays an important role in abiotic stress tolerance; in particular, its C-terminal structural domains interact with other proteins to identify substrates [56]. An F-box gene, *TaFBA1*, was identified in wheat organs, including roots [57]. In other few studies, overexpression of the gene showed drought tolerance, oxidative stress tolerance, heat tolerance, and longer root lengths relative to wild-genotype transgenic tobacco and *Arabidopsis* [56,57,58,59]. In rice, overexpression of *MAIF1* improved abiotic stress tolerance and helped to control root growth through multiple signal pathways, including negative signaling with abscisic acid (ABA), auxin, and ABA interactions, and the sucrose signaling pathway [62]. In *Arabidopsis*, the F-box protein *At1g08710* was found in roots, with longer roots under drought stress, and thus showed drought tolerance [60], whereas the F-box protein *SKP2B* (S-phase kinase-associated protein 2B) reduced lateral root formation by suppressing cell division in meristematic and founder cells [70]. In soybean, *GmFBX176* increased root length to tolerate salt stress [61]. A meta-QTL analysis found that *TraesCS7A01G481200* on chromosome 7A regulated root mass by encoding F-box protein [27].

*TraesCS7A02G431500* identified in pairs 9–10 encode protein kinase. Alpha4 (α4) subunit of casein kinase 2, CK2 α4, a protein kinase regulated primary root elongation, lateral root formation, and root development in *Arabidopsis* [63,66]. Overexpression of *stress/ABA-activated protein kinase 10* (*SAPK10*) in transgenic rice produced longer root hairs than the wild type [71]. In soybean, a root-specific serine/threonine kinase WNK (with no lysine K) homolog, *GmWNK1*, played an important role in root system architecture; overexpression of *35S-GmWNK1* in transgenic soybean produced shallower and fewer lateral roots than the wild type [67]. Furthermore, overexpression of *GmWNK1* improved salt stress tolerance in *Arabidopsis*, possibly due to improved root traits [72]. In rice, *phosphorus-starvation tolerance 1* (*PSTOL1*), a protein kinase gene, played an important role in improving root traits, including early growth of primary roots and increased rooting depth, root mass, and surface area; consequently, the gene improved N and P uptake [64]. In wheat, besides serine/threonine protein kinase genes, other protein kinase genes, such as *TaSnRK2.4*, *TaCIPK29*, and *TaMPK4*, affected root traits (including root length, root volume, and root surface area), grain yield, and abiotic stress tolerance [73,74,75]. Interestingly, an ortholog of an abiotic stress (drought and salt)-tolerant gene (*OsMAPK5*) in rice was found in wheat root tissue, indicating the importance of the protein-coding gene in abiotic stress tolerance [65]. Despite many protein kinases reports on different root traits in wheat, none has associated the protein with root mass. However, our study suggests that protein kinase-encoding genes contribute to genetic variations in root mass. Therefore, there is great scope for using the genes to improve root mass and other traits.

The seven NIL pairs significantly differed between the isolines for 11 important root traits. We identified 15 important genes on chromosomes 4BS, 4AS, and 7AL that encode proteins involved in controlling root traits. Of these, several have functions frequently reported as important proteins functioning in the controlling root traits of crops other than wheat, such as *TraesCS4B01G114500* encoding the Phox-associated domain, *TraesCS4A02G185300* and *TraesCS4A02G442700* encoding UDP-glycosyltransferase (UGT), and *TraesCS4A02G330900* encoding leucine-rich repeat receptor-like protein kinases (LRR-RLK). These candidate genes are important for further functional validation studies and marker developments to improve root traits in wheat.

## 4. Materials and Methods

### 4.1. Plant Material

Ten pairs of wheat near-isogenic lines (NILs) were used in this study (Table 1). The NIL pairs were developed using the heterogeneous inbred family (HIF) method and fast generation cycling system [17]. Briefly, from F2 progenies of a parental cross, heterozygous plants were selected using linked markers and selfed to F8 generation, and the advance of the generations followed a single seed descent (SSD) method [17,19,20]. Pairs 1–4 were developed from Dharwar Dry × C306, targeting a drought-tolerance-controlling GR (qDSI.4B.1) on chromosome 4BS [17]; pairs 5–7 were developed from Babax × Dharwar Dry targeting a drought-tolerance-controlling GR (qDT.4A.1) on chromosome 4AS [18]; and pair 8 was developed from Chara × DN5637B*8 targeting a preharvest sprouting-tolerance-controlling GR (QPhs.ocs-4B.1) on chromosome 4BL. Pairs 9–10 were developed from Cascades × W156 targeting heat-tolerance-controlling GR (QHtscc.ksu-7A) on chromosome 4AL [20].

### 4.2. Phenotyping for Root Traits

A randomized block design was used in a semi-hydroponic phenotyping system [76] for the 10 NIL pairs with three replications. The experiment was conducted in a temperature-controlled (10–24 °C) glasshouse at the University of Western Australia, Perth, from mid-June to late August 2019. The semi-hydroponic system included a 240 L (751 × 850 mm top, 1080 mm depth) mobile plastic bin, accommodating 16 growth units made from acrylic panels (260 mm wide × 480 mm long × 5 mm thick) wrapped in black calico cloth. The design, assembly, and operation of the semi-hydroponic system are described in Chen et al. (2011, 2020) [16,76]. The bin was filled with 35 L of nutrient solution (in mM): N (2000), K (1220), P (20), S (1802), Na (0.06), Ca (600), Mg (200), Cu (0.2), Zn (0.75), Mn (0.75), B (5), Co (0.2), Mo (0.03), and Fe (20) [16,76]. Wheat seeds were sterilized with 1% sodium hypochlorite (NaOCl) and germinated in multiple-welled plastic trays containing washed river sand (<2 mm). Seedlings with 4−5 cm long roots were washed in deionized (DI) water and transplanted into the bins. The plants in the growth units were watered with a controlled irrigation system described in Chen et al. (2020) [16]. Based on the evapotranspiration rate calculated during the trial experiment, 7 L of nutrient solution was added to each bin weekly. The position of each bin was rotated weekly to minimize environmental effects.

### 4.3. Trait Measurements

All plants were harvested 42 days after transplanting at tiller onset (Zadoks 2.4) [77] when the deeper root systems reached the surface of the nutrient solution. Leaf number per plant (NLP) was counted the day before harvest. At harvest, each growth panel was removed from the bin and laid out on a flat bench. After removing the wrapped cloth, maximal shoot height (SH) and rooting depth (RD; maximum root depth of an individual plant measured from the crown) were measured with a ruler, and the number of nodal roots (NNR) counted. A portable photographing system, including a fluorescence light and Nikon D5100 camera, was set up above the bench [78] to photograph the root systems. After photographing, the roots were separated from the shoots at the crown, and the number of nodal roots counted. Shoots were cut from the crown and dried in an air-forced oven at 65 °C for 72 h to obtain shoot dry weight. Each root system was cut into 20 cm sections for optical scanning at 400 dpi using a desktop scanner (Epson Perfection V800/850). Root morphological traits (total root length (RL), root diameter (Rdia), root surface areas (RSAs), and root diameter length (RDCL) were determined for each root section after analyzing the root images in WinRHIZO Pro software (v2009, Regent Instruments Inc., Montreal, QC, Canada) [16]. The total root length (RL) was the sum of all root length sections from the same plant. Root diameter length and surface areas in two root diameter classes (0–0.25 mm and >0.25 mm) were measured. All root sections were combined for the same plant and dried in an air-forced oven at 65 °C for 72 h to obtain root dry mass (RM). Root/shoot ratio (RSR) was the ratio of RM and shoot dry mass. Specific root length (SRL) was the total root length per unit of biomass.

### 4.4. Statistics

A two-sample Mann–Whitney U-test was conducted using R package (R version 4.0.3) to detect significant differences between the isolines. A correlation analysis was done using RStudio Version 1.2.5033.

### 4.5. 90K SNP Genotyping and Candidate Gene Identification

The wheat 90K Illumina iSelect array containing 81,587 SNPs was used to genotype genomic DNA samples of the NILs showing contrasting phenotypes [6]. SNP genotype calling and clustering were done with GenomeStudio 2.0 software (San Diego, California, United States) (Illumina, https://www.illumina.com, accessed on 2 March 2021). SNPs with no polymorphism, or those with more than 20% missing values, minor allele frequency (MAF) <0.05, or heterozygous calls >0.25 were deleted. The polymorphic SNP loci between each NIL pair were then analyzed to identify candidate genes associated with root traits.

The sequence of each polymorphic SNP was identified from the Triticeae Toolbox (T3) (https://triticeaetoolbox.org/wheat/, accessed on 2 March 2021) database and blasted with the wheat reference genome RefSeq v1.0 to locate the physical position [79]. To identify candidate genes, JBrowse (http://www.wheatgenome.org/Tools-and-Resources/Sequences, accessed on 2 March 2021) databasewas used. Genes closely associated with the screening markers of the targeted QTL region and those genes located up to two mega base pairs (Mb) away from the SNPs [80] were scrutinized to identify candidate genes. The functions of the candidate high-confidence genes were searched on the International Wheat Genome Sequencing Consortium (IWGSC) RefSeq v1.0 website (https://wheat-urgi.versailles.inra.fr/Seq-Repository/Annotation, accessed on 2 March 2021) [15]. Genes previously related to the growth and development of root traits in plants were considered potential candidate genes.

## Figures and Tables

**Figure 1 ijms-22-03579-f001:**
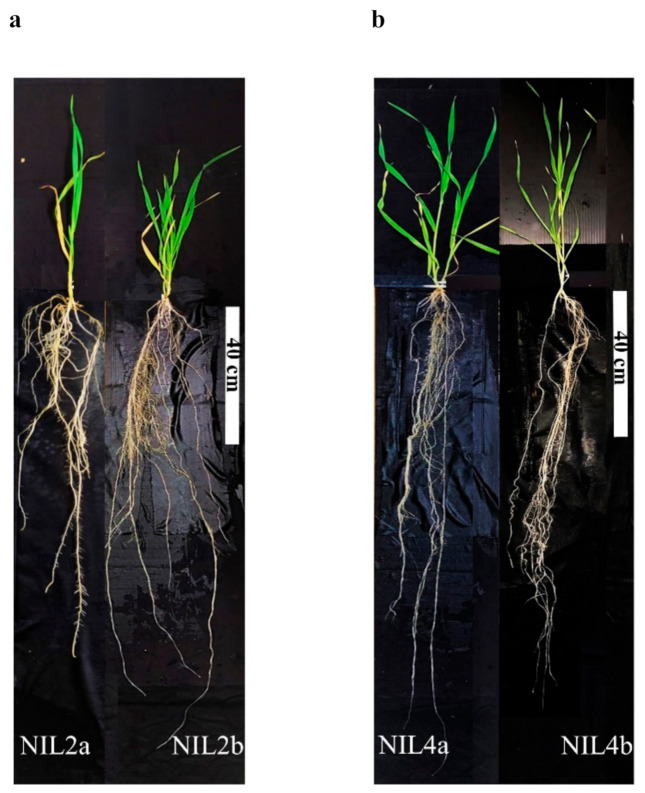
Contrasting rooting depth of two pairs of wheat near-isogenic lines (NILs) at 42 days after transplanting in a semi-hydroponic system in a glasshouse: (**a**) rooting depth of a NIL2a (72.8 cm) and a NIL2b (134.2 cm) plant, and (**b**) a NIL4a (134 cm) and a NIL4b (95.4 cm) plant. The white bar indicates a 40 cm scale.

**Figure 2 ijms-22-03579-f002:**
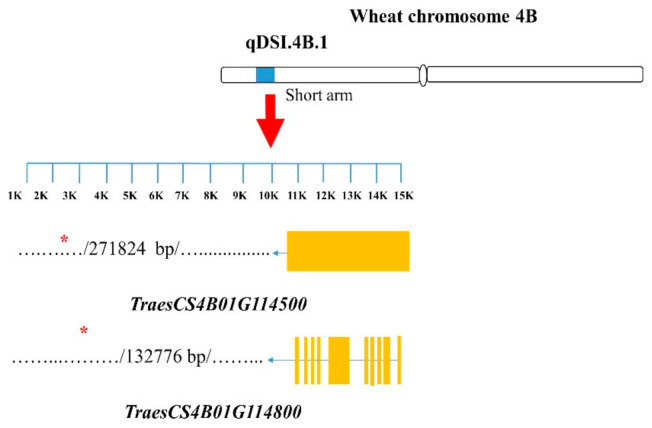
Structures of candidate genes in the qDSI.4B.1 genomic region (GR). SNP markers are located in different positions of the genes. The genes were identified based on the 90K wheat microarray assessment of near-isogenic lines with contrasting root traits. SNP markers are located in different positions of the genes. Structural information of the genes and SNP markers were obtained from a wheat genome database (https://urgi.versailles.inra.fr/jbrowseiwgsc/gmod_jbrowse/, accessed on 2 March 2021). The measuring bar indicates the DNA length in kilo base pairs (kb).

**Figure 3 ijms-22-03579-f003:**
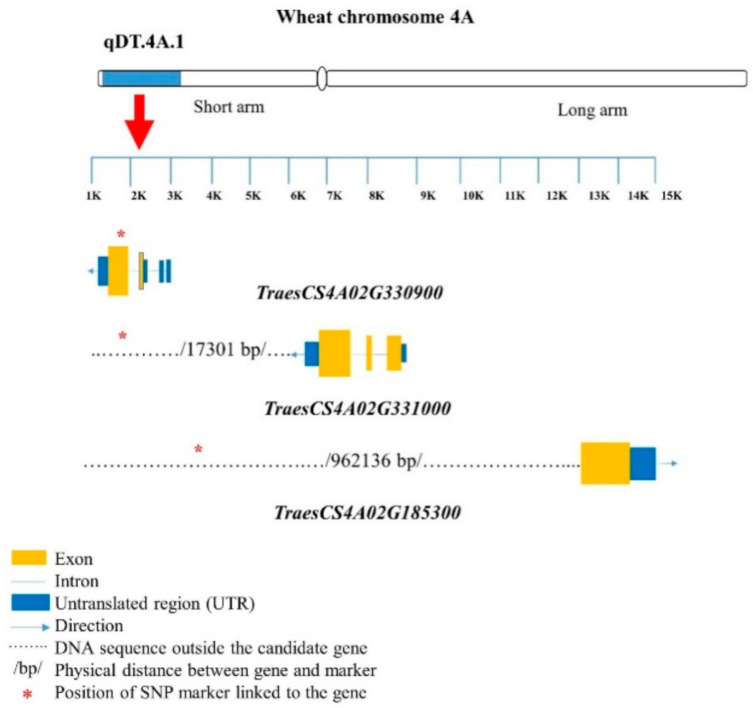
Structures of candidate genes in the qDT.4A.1 genomic region (GR). SNP markers are located in different positions of the genes. The genes were identified based on the 90K wheat microarray assessment of near-isogenic lines with contrasting root traits. SNP markers are located in different positions of the genes. Structural information of the genes and SNP markers were obtained from wheat genome database (https://urgi.versailles.inra.fr/jbrowseiwgsc/gmod_jbrowse/) (accessed on 2 March 2021). The measuring bar indicates the DNA length in kilo base pairs (kb).

**Figure 4 ijms-22-03579-f004:**
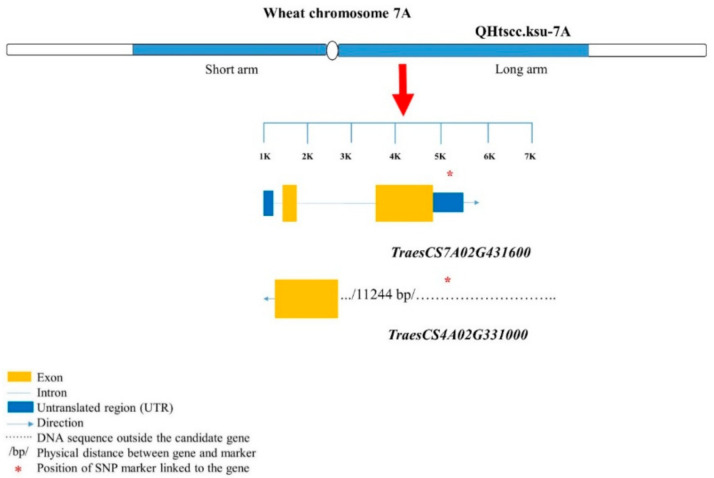
Structures of candidate genes in the QHtscc.ksu-7A genomic region (GR). The genes were identified based on the 90K wheat microarray assessment of near-isogenic lines with contrasting root traits. SNP markers are located in different positions of the genes. Structural information of the genes and SNPs was from a wheat genome database (https://urgi.versailles.inra.fr/jbrowseiwgsc/gmod_jbrowse/, accessed on 2 March 2021). The physical positions of the linking markers of the QTL were found from the original mapping study using parental cultivar; therefore, in the reference genome (Chinese Spring), the QTL extends in both the long and short arm of chromosome 7A, which is unusual. Chromosomal fragment translocation might be the possible reason for the difference between the reference genome and the parental cultivars used in the original mapping study [25]. The measuring bar indicates the DNA length in kilo base pairs (kb).

**Table 1 ijms-22-03579-t001:** Background information on the 10 pairs of wheat near-isogenic lines (NILs) used in this study.

Pair No.	NIL Names	Original NIL Name	Parents ^#^	Targeted Traits to Develop NIL	Targeted QTL Name	Chromosomes	Marker Intervals of Target QTL ^†^	Physical Positions (Mb) of Target QTL	References
1	NIL1a	qDSI.4B.1-1	Dharwar Dry × **C306**	Drought tolerance	qDSI.4B.1	4BS	**gwm368**-barc20	59.61–75.74	[17,21]
	NIL1b	qDSI.4B.1-1							
2	NIL2a	qDSI.4B.1-2							
	NIL2b	qDSI.4B.1-2							
3	NIL3a	qDSI.4B.1-3							
	NIL3b	qDSI.4B.1-3							
4	NIL4a	qDSI.4B.1-4							
	NIL4b	qDSI.4B.1-4							
5	NIL5a	BD NIL-1	Babax × **Dharwar Dry**	Drought tolerance	qDT.4A.1	4AS	Xgwm397-**Xwmc491**	0.14–94.02	[18]
	NIL5b	BD NIL-1							
6	NIL6a	BD NIL-3							
	NIL6b	BD NIL-3							
7	NIL7a	BD NIL-4							
	NIL7b	BD NIL-4							
8	NIL8a	NIL-PHSR4BL-6R	Chara × **DN5637B*8**	Preharvest sprouting resistance	QPhs.ocs-4B.1	4BL	**Xgwm495**-Xgwm375	482.82–568.56	[19,22]
	NIL8b	NIL-PHSR4BL-6S							
9	NIL9a	NIL 9 (+)	**Cascades** × W156	Heat tolerance	QHtscc.ksu-7A	7AL	**Xbarc49**-Xbarc121	155.05–611.84	[20,23]
	NIL9b	NIL 9 (−)							
10	NIL10a	NIL 10 (+)							
	NIL10b	NIL 10 (-)							

^#^ Bold indicates donor parent of the positive allele of respective quantitative trait loci (QTL). ^†^ Bold indicates screening marker used for developing the NILs.

**Table 2 ijms-22-03579-t002:** Phenotypic performances (mean ± standard error) of wheat near-isogenic lines (NILs) used in this study.

NIL Pairs			Root Traits									Shoot Traits	
	RL (cm)	RM (g)	RD (cm)	RSR	RSA1 (cm^2^)	RSA2 (cm^2^)	Rdia (mm)	RDCL1 (mm)	RDCL2 (mm)	SRL (cm g^−1^)	NNR	SH (cm)	NLP
1	1890.00 ± 248.70	0.15 ± 0.02	54.00 ± 16.84	0.46 ± 0.05	51.87 ± 10.79	12.35 ± 1.80	0.40 ± 0.05	306.13 ± 51.31	337.25 ± 61.04	12,653.00 ± 1379.00	5.00 ± 1.52	34.73 ± 3.42	**7.00 ± 0.27 ***
	2571.00 ± *889.51*	0.20 ± 0.05	75.43 ± 16.44	0.44 ± 0.02	37.81 ± 2.77	12.73 ± 3.95	0.48 ± 0.08	294.94 ± 88.37	240.67 ± 31.44	11,506.00 ± 1701.00	4.00 ± 0.72	34.97 ± 4.89	9.00 ± 0.27
2	3407.00 ± *948.04*	0.29 ± 0.08	86.57 ± 15.80	0.371 ± 0.04	31.60 ± 4.68	17.41 ± 2.44	0.34 ± 0.01	436.00 ± 56.58	213.79 ± 34.34	11,757.00 ± 163.44	6.00 ± 1.44	34.63 ± 2.07	12.00 ± 1.63
	6505.00 ± *657.71*	0.41 ± 0.05	110.90 ± 15.80	0.43 ± 0.01	47.40 ± 11.32	23.69 ± 1.73	0.36 ± 0.02	**607.75 ± 29.76 ***	299.24 ± 67.96	**15,954.00 ± 727.89 ***	7.00 ± 0.54	**44.63 ± 1.74 ***	12.00 ± 1.25
3	3392.00 ± *924.69*	0.23 ± 0.05	82.5 ± 14.99	**0.46 ± 0.05 ***	49.64 ± 2.30	17.82 ± 2.33	0.37 ± 0.01	421.35 ± 59.61	324.45 ± 5.20	14,082.00 ± 937.50	5.00 ± 0.47	28.47 ± 3.61	8.00 ± 1.19
	2770.00 ± 45.03	0.14 ± 0.04	66.13 ± 14.93	0.20 ± 0.04	58.70 ± 10.17	13.62 ± 2.87	0.39 ± 0.03	324.24 ± 70.50	372.32 ± 53.80	25,085.00 ± 7532.00	5.00 ± 0.98	39.20 ± 3.84	8.00 ± 0.54
4	6668.00 ± 987.52	0.39 ± 0.04	106.87 ± 15.82	0.41 ± 0.01	**68.35 ± 6.71 ***	**32.48 ± 3.36 ***	0.35 ± 0.01	**688.57 ± 40.15 ***	**473.52 ± 43.44 ***	16,860.00 ± 1175.00	6.00 ± 0.47	50.27 ± 1.55	10.00 ± 0.72
	4639.00 ± 188.52	0.29 ± 0.01	92.87.13 ± 14.59	0.39 ± 0.02	46.21 ± 5.18	20.89 ± 1.12	0.33 ± 0.01	503.65 ± 33.73	316.00 ± 41.25	16,336.00 ± 1229.00	5.00 ± 0.27	51.40 ± 1.16	9.00 ± 0.72
5	3516.00 ± 432.68	0.24 ± 0.05	51.37 ± 14.85	**0.40 ± 0.01 ***	84.19 ± 12.02	26.23 ± 3.66	0.40 ± 0.01	626.27 ± 93.21	680.17 ± 154.77	15,753.00 ± 1840.00	4.00 ± 0.72	36.50 ± 1.78	12.00 ± 0.27
	2283.00 ± 321.92	0.12 ± 0.02	50.03 ± 15.29	0.32 ± 0.01	65.67 ± 10.37	18.83 ± 3.52	0.38 ± 0.03	414.93 ± 88.54	460.85 ± 72.67	19,807.00 ± 1074.00	3.00 ± 0.27	36.53 ± 2.69	10.00 ± 1.19
6	**6850.00 ± 1230.32 ***	**0.38 ± 0.08 ***	79.20 ± 15.01	0.85 ± 0.29	48.24 ± 3.03	**33.71 ± 3.46 ***	0.35 ± 0.03	**838.56 ± 92.89 ***	341.37 ± 26.16	18,281.00 ± 868.08	6.00 ± 1.44	38.80 ± 1.27	**12.00 ± 0 ***
	2383.00 ± 349.25	0.10 ± 0.01	45.13 ± 15.13	0.31 ± 0.01	57.40 ± 7.92	16.86 ± 2.34	0.41 ± 0.01	360.57 ± 51.17	433.71 ± 69.77	**24,781.00 ± 690.00 ***	2.00 ± 0.47	36.10 ± 1.21	9.00 ± 0.82
7	4042.00 ± 1284.63	0.20 ± 0.05	49.77 ± 15.43	0.45 ± 0.06	73.12 ± 22.21	23.48 ± 6.02	0.38 ± 0.01	537.60 ± 115.42	498.18 ± 148.72	18,556.00 ± 2003.00	3.00 ± 0.54	35.20 ± 3.59	11.00 ± 2.60
	3683.00 ± 183.32	0.25 ± 0.03	47.67 ± 15.87	0.49 ± 0.02	82.49 ± 12.10	22.64 ± 0.58	0.42 ± 0.04	529.10 ± 49.73	585.02 ± 93.87	15,004.00 ± 1463.00	3.00 ± 0.47	36.23 ± 0.98	11.00 ± 1.70
8	5436.00 ± 221.17	0.26 ± 0.01	95.07 ± 15.87	0.47 ± 0.01	46.80 ± 14.10	27.95 ± 5.15	0.33 ± 0.01	657.88 ± 98.61	334.18 ± 111.49	20,826.00 ± 2672.00	5.00 ± 0.27	32.60 ± 1.18	13.00 ± 0.72
	7379.00 ± 498.55	0.24 ± 0.02	74.90 ± 16.96	0.42 ± 0.06	67.35 ± 8.80	38.10 ± 2.93	0.35 ± 0.02	789.38 ± 63.83	612.30 ± 46.61	31,113.00 ± 1872.00	4.00 ± 0.72	37.83 ± 3.30	14.00 ± 1.52
9	3181.00 ± 845.11	0.14 ± 0.02	87.67 ± 15.51	0.58 ± 0.21	38.38 ± 7.40	19.27 ± 1.73	0.34 ± 0.01	**439.01 ± 26.44 ***	292.08 ± 55.41	24,687.00 ± 3682.00	2.00 ± 0.54	32.30 ± 2.75	9.00 ± 0.47
	1404.00± 773.75	0.06 ± 0.01	44.43 ± 16.81	0.28 ± 0.03	34.78 ± 14.22	10.93 ± 2.69	0.37 ± 0.03	255.65 ± 48.74	257.46 ± 106.02	23,379.00 ± 2924.00	1.00 ± 0.47	28.97 ± 4.41	7.00 ± 1.70
10	**3331.00 ± 289.13 ***	0.16 ± 0.04	59.20 ± 16.38	0.34 ± 0.05	62.45 ± 2.04	21.17 ± 2.75	**0.43 ± 0.03 ***	450.23 ± 58.75	**466.33 ± 08.23 ***	23,364.00 ± 4085.00	3.00 ± 0.27	41.90 ± 6.80	10.00 ± 0.72
	5008.00 ± 498.33	0.29 ± 0.05	95.63 ± 0.00	0.47 ± 0.05	41.86 ± 7.33	27.18 ± 3.63	0.31 ± 0.02	680.67 ± 87.33	289.04 ± 55.34	18,221.00 ± 1564.00	**5.00 ± 0.27 ***	39.57 ± 1.26	13.00 ± 0.72

RL = total root length, RM = root dry mass, RD = rooting depth, RSR = root/shoot ratio, RSA1 = root surface area (>0.25 mm diameter class), RSA2 = root surface area (<0.25 mm diameter class), RDia = root diameter, RDCL1 = root diameter class length (<0.25 mm), RDCL2 = root diameter class length (>0.25 mm), SRL = specific root length, NNR = nodal root number per plant, SH = shoot height, NLP = leaf number per plant. “*” indicates significant difference at 5% level of significance at nonparametric Mann–Whitney U-test. Bold indicates significant difference between the isolines.

**Table 3 ijms-22-03579-t003:** Candidate genes identified in the targeted QTL regions are strongly suggested to be responsible for the tested root and shoot traits in the nine pairs of wheat near-isogenic lines (NILs) used in this study.

NIL Pairs	Traits with Significant Differences between Isolines	SNP	Candidate Gene	Gene Function	Marker–Gene Distance
1–4	RSR, RSA1, RSA2, RDCL1, RDCL2, SRL, SH, and NLP	Excalibur_c869_2091	*TraesCS4B01G114500*	Phox-associated domain, sorting nexin isoform 3	271.82 kb away
			*TraesCS4B01G114800*	bHLH-MYC and R2R3-MYB transcription factors N-terminal	132.78 kb away
5–7	RL, RM, RSR, RSA2, RDCL1, SRL, and NLP	BobWhite_c22126_94	*TraesCS4A02G163700*	Inositol-1-monophosphatase family protein	1064.77 kb away
		BobWhite_c27287_232	*TraesCS4A02G330900*	LRR-RLK	7.78 kb away
			*TraesCS4A02G331000*	F-box family protein	17.30 kb away
		BS00098868_51	*TraesCS4A02G215800*		202.63 kb away
			*TraesCS4A02G215700*		214.39 kb away
		BS00094406_51	*TraesCS4A02G185300*	UDP-glucoronosyl and/or UDP-glucosyl transferase	962.14 kb away
		CAP12_c3789_390	*TraesCS4A02G442700*		11.58 kb away
		RAC875_c51781_771			10.29 kb away
		Tdurum_contig12899_342	*TraesCS4A02G323500*	Protein kinase; serine–threonine/tyrosine- protein kinase	5.86 kb away
			*TraesCS4A02G323400*		10.11 kb away
		Jagger_c9935_144	*TraesCS4A02G000300*		90.30 kb away
		Excalibur_c94546_61	*TraesCS7A02G101000*	GH3 family, GH3 auxin-responsive promoter	31.67 kb away
9–10	RL, Rdia, RDCL1, RDCL2, and NNR	CAP8_c3496_118	*TraesCS7A02G431600*	F-box domain	Overlapped in UTR
			*TraesCS7A02G431500*	Protein kinase; serine–threonine/tyrosine- protein kinase	11.24 kb away

RL = total root length, RM = root dry mass, RSR = root/shoot ratio, RSA1 = root surface area (>0.25 mm diameter class), RSA2 = root surface area (<0.25 mm diameter class), RDia = root diameter, RDCL1 = root diameter class length (<0.25 mm), RDCL2 = root diameter class length (>0.25 mm), SRL = specific root length, NNR = nodal root number per plant, SH = shoot height, NLP = leaf number per plant; SNP = single nucleotide polymorphism

**Table 4 ijms-22-03579-t004:** Proteins and genes identified as responsible for root traits in crops and *Arabidopsis*.

Protein Name	Gene Name	Reported Root Trait	References
Phox-associated	*TraesCS4B01G114500*	Root growth	[43,44]
domain, sorting nexin		development and root	
isoform 3		hair growth in *Arabidopsis*	
bHLH-MYC and R2R3-MYB transcription factors N-terminal	*TraesCS4B01G114800*	Root growth elongation; root hair growth in *Arabidopsis*, rice; abiotic stress tolerance in wheat	[46,47,48,49,50,51,52,54]
Inositol-1-monophosphatase family protein	*TraesCS4A02G163700*	Rooting depth initiation and elongation in *Arabidopsis*	[55]
LRR-RLK	*TraesCS4A02G330900*	Root length and root hair growth in *Arabidopsis*	[41,42]
F-box family protein	*TraesCS4A02G331000* *TraesCS4A02G215800* *TraesCS4A02G215700* *TraesCS7A02G431600*	Root length in *Arabidopsis*, rice, and soybean; abiotic stress tolerance in different crops	[27,56,57,58,59,60,61,62]
UDP-glucoronosyl and/or UDP-glucosyl transferase	*TraesCS4A02G185300* *TraesCS4A02G442700*	Root length and root mass in rice and canola	[38,39]
Protein kinase; serine–threonine/tyrosine- protein kinase	*TraesCS4B01G207500* *TraesCS4B01G299900* *TraesCS4B01G210600* *TraesCS4B01G130700* *TraesCS4B01G238700* *TraesCS7A02G431500*	Root length, root number, root growth in *Arabidopsis*, rice, and soybean; nutrient stress tolerance	[63,64,65,66,67]
GH3 family, GH3 auxin-responsive promoter	*TraesCS7A02G101000*	Root growth, development, root length in rice, *Arabidopsis*; abiotic stress tolerance	[68,69]

## Data Availability

Not applicable.

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
