# Peer review of "Identification of Candidate Genes for Root Traits Using Genotype–Phenotype Association Analysis of Near-Isogenic Lines in Hexaploid Wheat (Triticum aestivum L.)"

_ijms, 2021, doi:10.3390/ijms22073579_

Round 1
Reviewer 1 Report
This article is interesting with new data and results which might shift the wheat breeding to create more suitable varieties for the changing climate. The architecture of the wheat root system is the key trait to fight against abiotic stresses. Although there are some sentences that need to be reworded (e.g. line 237) and the statistical software used to detect significant differences between isolines does not seem appropriate to me for this type of article, I recommend it for publication. It would help if the authors are able to improve the style and use other statistical software.
Author Response
Response:
We thank the reviewer for the useful suggestions. The sentence has been modified to make it clearer: “Besides the root trait QTL, a few other yield-related QTL, such as QTL for thousand kernel weight, were also found on chromosome 4BS as reported earlier [26, 30]” (L269). For the statistical software used to detect significant differences between isolines, we have redone the statistical analysis (which was also suggested by Reviewer 2) using a non-parametric test (Mann-Whitney U test in R package) in addition to t-test. The Abstract, Materials and Methods, and Results sections, Table 2 and Supplementary Figure 1 were revised to reflect the above changes.
Reviewer 2 Report
The manuscript entitled “Identification of candidate genes for root traits using genotype–phenotype association analysis of near-isogenic lines in hexaploid wheat (Triticum aestivum L)” submitted to IJMS, aims to identify candidate genes associated with root traits by using NILs and previously reported information on genomic regions.
COMMENTS TO THE AUTHORS
Material and methods:
L451 please check reference format
The abbreviations of traits used in the manuscript should be included in the traits measurements description
Statistics: Do you think a t-test is the most suitable statistics in a design with only 3 replicates? With such a low sample size, I would use a non-parametric test more adequate for these situations. Do you think enough the number of replicates for this kind of experiment? Please justify
Results:
2.1. Table 2 should be changed, it is quite difficult to see the relevant information. There is no need to show all the p values, the notation with *, **,** for p values <0.05, 0.01 and 0.001 is enough. The pairs where significant differences have been found could be highlighted in bold or underlined. The authors could consider moving this table to supplementary material and only show in the manuscript a summary (table or figure), indicating only the mean and SD values when significant differences have been found. Please specify if the value after the mean corresponds to SD or to SE. I would suggest adding a table with the raw data for the three replicates as supplementary.
What is the correlation between characters?
Why did you not include the parent lines in the phenotypic analysis? Please, explain
In the supplementary figure, the error bars in total root length (a) don’t match with the table values. Moreover, it seems that the units in Table 2 and Figure S1 are wrong (60 meters root length??)
L88-89: “all NIL pairs significantly differed between isolines for the measured root traits”. With this sentence it seems that there are differences for most of the characters, when there are NIL pairs (pair 3 and 8) that differ in only one character, and pair 7 in no one. Please rewrite.
2.2. L133, It is said:“108 SNPs showed polymorphism for different root traits”. Does not make sense, please rewrite
It is not clear if all the NILs pairs derived from the same parental lines (i.e.1, 2, 3, and 4), are polymorphic for the same SNPs (11). Please, explain.
In table S1, the physical position of the target QTL should be in the same format as the physical position of SNPs. I don’t understand how are the SNPs ordered in the table, should be ordered by position. How many of them are in the QTL interval?
In Table 3, the SNP Excalibur_c869_2091 appears to be associated to candidate gene TraesCS4B01G114500, but this gene does not appear in Table S1. Please, revise carefully the data.
In my opinion, with the data obtained, authors cannot established that “Candidate genes identified in the targeted QTL regions responsible for the tested root and shoot traits in the ten pairs of wheat near-isogenic lines (NILs)”
In the pair of NILs 8, significant differences are found only for one of the root traits measured, and it is one (SRL) calculated from other traits. Is this really the best material for this analysis?
In Figures 2, 3, and 4, only some of the candidate genes are represented. How have they been selected? Please explain.
GENERAL COMMENT
I agree with the authors that this plant material, the group of Nils pairs, is quite interesting for genetic dissection of these specific genomic regions. However, with the phenotypic results presented here, I am not sure that they can confirm the association of these GR with root phenotype. The authors should explain much better the suitability of this plant material for the aim of the study. Why different NILs pairs from the same crossing behave so differently in relation to root/shoot traits? I.e. pair 6 is different for seven traits and in pair 7 no trait showed differences. From the text, I assume that they are only 50 SNPs polymorphic between these pairs of lines, the same between both pairs, and all of them in the QTL target region, so, how do you explain that??
From my point of view, all the genomic analysis could be of interest, but it is completely independent of the phenotype analysis. It is a simple description of candidate genes inside the GR from previously published studies and genome databases. I am sorry but I cannot consider this manuscript good enough to be published in the IJMS.
Author Response
- I agree with the authors that this plant material, the group of NIL pairs, is quite interesting for genetic dissection of these specific genomic regions. However, with the phenotypic results presented here, I am not sure that they can confirm the association of these GR with root phenotype. The authors should explain much better the suitability of this plant material for the aim of the study. Why different NIL pairs from the same crossing behave so differently in relation to root/shoot traits? I.e. pair 6 is different for seven traits and in pair 7 no trait showed differences. From the text, I assume that they are only 50 SNPs polymorphic between these pairs of lines, the same between both pairs, and all of them in the QTL target region, so, how do you explain that?
From my point of view, all the genomic analysis could be of interest, but it is completely independent of the phenotype analysis. It is a simple description of candidate genes inside the GR from previously published studies and genome databases. I am sorry but I cannot consider this manuscript good enough to be published in the IJMS.
Response:
We appreciate that the reviewer has recognized the importance of NILs as the plant materials used in the study. We believe that some background information about the NILs would help to address the above comments. The NILs used in this study were developed using heterogeneous inbred family (HIF) method, which ensured that ~99.3% of genetic backgrounds of the isolines in each pair are the same. For each targeted genomic region, we used a QTL flanking marker to screen the lines in each generation during the NIL development, therefore the QTL region is specially targeted in the NIL population. For the different NIL pairs from the same cross, apart from the targeted QTL region, each NIL pair will also have a different genetic background from other pairs (as each NIL pair is developed from a single seed descend of an F2 individual of the cross progeny, and the F2 individuals’ genetic backgrounds are different from each other) --- the NIL development procedure has been described in our previous publications (Wang et al., 2019; Mia et al., 2019; Lu et al., 2020).
That is why different NIL pairs from the same cross can behave so differently in relation to root/shoot traits. The difference among different NILs is due to the genetic backgrounds while the difference between the isolines of each NIL pairs is due to the difference of the target GR. Therefore, idetification of the common candidate genes among NILs with different genetic backgrounds targeting the same genomic region will be useful; it will significantly eliminate the noises and discover the real candidates for the root traits which showed common performances in different pairs. Revelant background information has been added in the Introduction, Materials and Methods and Discussion section.
Below is the clear explanation of the answers in different sections in the text:
In L79-86 of Introduction: The NILs used in the present study were developed targeting different genomic regions (GRs) on wheat chromosome arms 4BS [17], 4AS [18], 4BL [19], and 7AL [20]. The NILs were developed using heterogeneous family (HIF) method [19] and therefore, the genetic backgrounds among each pair are different even if they target the same GR, which allowed comparisons among the NIL pairs to narrow down the candidates of the trait-linked markers and genes. In this study, we investigated root traits in these ten pairs of wheat NILs, using the wheat 90K Infinium iSelect SNP array as a genotyping method.
In the Materials and Methods section: In L434-437: Briefly, from F2 progenies of a parental cross, heterozygous plants were selected using linked markers, and selfed to F8 generation, and the advance of the generations followed a single seed decent (SSD) method [17, 19, 20].
In discussion a new paragraph have been added (L234-249) as below:
3.1. NIL pairs show different phenotypic performances
Among the ten NIL pairs, five pairs showed significant differences between the isolines for the studied root traits. The HIF method for NIL development allowed the isolines of a NIL pair to be homozygous apart from the target locus [20]. Therefore, the isolines showed contrasting performances suggest they contain the gene(s) controlling the corresponding root traits. NIL pairs that were developed from the same parents showed different phenotypic performances among the pairs. Due to use of HIF method in NIL development, each NIL pair was from a single seed descent (SSD) of an F2 individuals, and for that reason [17] the NIL pairs had completely different genetic backgrounds from each other. Therefore, despite having the same parents, individual pairs showed different root phenotypic performances. By comparison among all the pairs targeting the same GR, we narrowed down in each GR for the candiate SNPs that could be linked to the root traits.
- L451 please check reference format
Response:
The reference (L454-456) has been formatted according to the journal guidelines.
- The abbreviations of traits used in the manuscript should be included in the traits measurements description
Response:
The abbreviations of traits have been included in the “Trait measurements” part of the Materials and Methods section.
- Statistics: Do you think a t-test is the most suitable statistics in a design with only 3 replicates? With such a low sample size, I would use a non-parametric test more adequate for these situations. Do you think enough the number of replicates for this kind of experiment? Please justify.
Response:
We agree with the above comment. We have redone the statistical analysis (which was also suggested by Reviewer 1) using a non-parametric test (Mann-Whitney U test in R package) in addition to t-test. This is now included in the revised Materials and Methods (L496-497) and Results sections (L94-98; L104-106 and L109) and Table 2 (L117-125).
- Table 2 should be changed, it is quite difficult to see the relevant information. There is no need to show all the p values, the notation with *, **,** for p values <0.05, 0.01 and 0.001 is enough. The pairs where significant differences have been found could be highlighted in bold or underlined. The authors could consider moving this table to supplementary material and only show in the manuscript a summary (table or figure), indicating only the mean and SD values when significant differences have been found. Please specify if the value after the mean corresponds to SD or to SE. I would suggest adding a table with the raw data for the three replicates as supplementary.
Response
Table 2 has been revised according to the suggestions: “p” value has been removed from the Table, and statistically significant pairs are in bold. The SD values corresponding to means have been changed into SE, and clearly indicated in the table title (L117). The raw data with three replicates are now included as a supplementary Table (Supplementary Table S1).
- What is the correlation between characters?
Response
A correlation analysis using “RStudio Version 1.2.5033” has been added in the Results section. A correlation matrix has been added as a supplementary figure (Figure S2). A sentence “A correlation analysis was done using RStudio Version 1.2.5033” was added in the “Statistics” section of Materials and Methods (L498).
The following paragraph has been added in the Results section under the “phenotypic variation, and correlation of root traits” as below (L133-146):
2.1 Phenotypic variation, and correlation of root traits
Significant correlation was found between different root and shoot traits of the NIL pairs. According to the correlation co-efficient values, strong (r= ≥0.7) and weak (r= ≤0.3) correlations were denoted [24]. Correlation coefficient values in between 0.3-0.7 were denoted as moderate correlation. Total root length showed strong positive correlation with RM, RSA2, RDCL1 and NLP, moderate positive correlation with RD, RSR, NNR, SH, and moderate but negative correlation with Rdia. Root mass showed strong positive correlation with RD, RDCL1 and NNR, and moderate positive correlation with RSR, RSA2, SH and NLP. Strong positive correlation was found between RD and NNR whereas moderate but negative correlation was found between RD and Rdia. Other traits such as RSA1 and RDCL2, RSA2 and RDCL1, RSA2 and NLP, and RDCL1 and NLP also showed strong positive correlation (Supplementary Figure S2).
- Why did you not include the parent lines in the phenotypic analysis? Please, explain
Response
This study aimed to characterize the NILs for elucidating the roles of the different targeted genomic regions on some root traits and we have considered the phenotypic differences between the isolines in each NILs. Phenotypic variations among the NIL pairs and among the parent lines were not considered, as they would serve little purpose for the research objectives as explained above (genetic background).
- In the supplementary figure, the error bars in total root length (a) don’t match with the table values. Moreover, it seems that the units in Table 2 and Figure S1 are wrong (60 meters root length??)
Response
In the Figure S1, total root length unit was in “meters” and also mentioned in the figure title. Total root length was more than 1000 cm (see Table 1). Therefore, to avoid the high value, metre was used as a unit instead of centimetre in the figure. Therefore, the standard errors also changed accordingly. The unit of the figures have been made bold for more visibility. A sentence “The bars indicate the standard errors” has been added in the figure legend to clarify.
- L88-89: “all measured root traits”. With this sentence it seems that there are differences for most of the characters, when there are NIL pairs (pair 3 and 8) that differ in only one character, and pair 7 in no one. Please rewrite.
Response
The sentence (L94) has been revised as “Except for Pair 1, Pair 7 and Pair 8, the NIL pairs significantly differed between isolines for the measured root traits”.
- 2.2. L133, It is said:“108 SNPs showed polymorphism for different root traits”. Does not make sense, please rewrite
Response
The sentence (L148) has been revised as “Of all the contrasting NIL pairs, a total of 67 SNPs on the targeted chromosomes were identified as candidate tightly linked markers for the studied root traits; for each investigated GR, they showed common polymorphism between the isolines among all the pairs targeting the same GR (Supplementary Table S2)”.
- It is not clear if all the NILs pairs derived from the same parental lines (i.e.1, 2, 3, and 4), are polymorphic for the same SNPs (11). Please, explain.
Response
The different NIL pairs derived from the same parents are not always polymorphic for the same SNPs, because because the genetic backgrounds among the pairs are different as we have explained above. Those common polymorphic SNPs in the different NIL pairs (from the same parents) were considered the most important SNPs for identifying the candidate genes. In Table S1, only these common polymorphic SNPs were shown. We have annotated in Table S1 to clarify this: “Only the common polymorphic SNPs among all the pairs targeting the same GRs are shown”. We have also modified the L148-160 of Results section as “Of all the contrasting NIL pairs, a total of 67 SNPs on the targeted chromosomes were identified as candidate tightly linked markers for the studied root traits; for each investigated GR, they showed common polymorphism between the isolines among all the pairs targeting the same GR (Supplementary Table S2); specifically, Pairs 1–4, Pairs 5–7 and Pairs 9–10 showed polymorphism for 11, 50 and six SNPs, respec-tively (Supplementary Table S2). In Pairs 1–4, Pairs 5–7 and Pairs 9–10, five, 11 and two SNPs showed polymorphism within the respective GRs. Using those 67 polymorphic SNP markers, from the nine NIL pairs, a total 181 candidate genes were identified (Supplementary Ta-ble S2). However, only those genes with functions that have been pre-viously reported to be responsible for root traits in wheat and other crops (especially cereal crops) were considered as important candidates (Table 3)” for clarification.
- In table S1, the physical position of the target QTL should be in the same format as the physical position of SNPs. I don’t understand how are the SNPs ordered in the table, should be ordered by position. How many of them are in the QTL interval?
Response
The QTL physical position has been changed into base pair (bp) from mega-base-pair (Mb) to add clarification. SNPs have been reorganised according to their physical positions. In Pairs 1-4, Pairs 5-7 and Pairs 9-10, five, 11 and two SNPs, respectively linked to genes contributing to root traits were in the QTL intervals, and a column indicating number of SNPs in the QTL interval has been added in the Table S1.
- In Table 3, the SNP Excalibur_c869_2091 appears to be associated to candidate gene TraesCS4B01G114500, but this gene does not appear in Table S1. Please, revise carefully the data.
Response
Excalibur_c869_2091 and related information have been added in the revised Table S1.
- In my opinion, with the data obtained, authors cannot established that “Candidate genes identified in the targeted QTL regions responsible for the tested root and shoot traits in the ten pairs of wheat near-isogenic lines (NILs)”
Response
We agree with the reviwer. We have modified our claim and the sentence (L167-168) has been changed to “Candidate genes identified in the targeted QTL regions are strongly suggested to be responsible for the tested root and shoot traits in the in the nine pairs of wheat near-isogenic lines (NILs) used in this study”.
- In the pair of NILs 8, significant differences are found only for one of the root traits measured, and it is one (SRL) calculated from other traits. Is this really the best material for this analysis?
Response
We agree with the reviewer. In our new non-parametric test, we have not found significant difference for any traits for Pair 8. Therefore, we have removed the SNPs and genes related to the pair from Table 3, Table 4, Supplementary Table 1 and Figure 2, and revised the text (Abstract, Results and Discussion) accordingly.
- In Figures 2, 3, and 4, only some of the candidate genes are represented. How have they been selected? Please explain.
Response
The genes were selected according to their annotated functions. Those with functions that have been previously reported to be responsible for root traits in wheat and other crops (especially cereal crops) were considered. Two sentences (L156-160) has been added in the “SNPs and candidate genes” point of Results section to clarify: “Using those 67 polymorphic SNP markers, from the entire nine NIL pairs, a total 181 candidate genes were identified (Supplementary Table S2). However, only those genes with functions that have been previously reported to be responsible for root traits in wheat and other crops (especially cereal crops) were considered as important candidates (Table 3)”.
Additional reference
- Kozak M, Krzanowski W, Tartanus M: Use of the correlation coefficient in agricultural sciences: problems, pitfalls and how to deal with them. Anais da Academia Brasileira de Ciências 2012, 84(4):1147-1156.
Round 2
Reviewer 2 Report
The authors have improved the manuscript and addressed all my questions. Now, I can consider it for publication.
Please, check the format of the supplementary tables in the final version. Table S2 is completely unintelligible.
Author Response
We thank the reviewer for the positive response on our revised manuscript. Supplementary Tables 1 & 2 are now reformatted as suggested to make it legible.
All documents are attached, we have removed the yellow highlights and this is the final version. We hope that the manuscript is acceptable to you.
Regards
Kadambot H.M. Siddique